# The Complex Role of Botulinum Toxin in Enhancing Goal Achievement for Post-Stroke Patients

**DOI:** 10.3390/toxins16040172

**Published:** 2024-03-31

**Authors:** Miruna Ioana Săndulescu, Delia Cinteză, Daniela Poenaru, Claudia-Gabriela Potcovaru, Horia Păunescu, Oana Andreia Coman

**Affiliations:** 1Doctoral School, Carol Davila University of Medicine and Pharmacy, 020021 Bucharest, Romania; daniela.poenaru@umfcd.ro (D.P.); horia.paunescu@umfcd.ro (H.P.); oana.coman@umfcd.ro (O.A.C.); 2Rehabilitation Department 9, Carol Davila University of Medicine and Pharmacy, 020021 Bucharest, Romania; 3Department of Pharmacology and Pharmacotherapy, Carol Davila University of Medicine and Pharmacy, 020021 Bucharest, Romania; 4National Institute of Rehabilitation, Physical Medicine and Balneo-Climatology, 010024 Bucharest, Romania

**Keywords:** abobotulinum-toxin-A, Goal-Attainment-Scale, stroke, spasticity, Modified-Ashworth-Scale, Numerical-Rating-Scale, passive-range-of-motion

## Abstract

Introduction. The rehabilitation medical team is responsible for the therapeutic management of post-stroke patients and, therefore, for the complex therapeutic approach of spasticity. Considering the generous arsenal at our disposal in terms of both pharmacological treatment, through the possibility of administering botulinum toxin to combat spasticity, and in terms of accurate assessment through developed functional scales such as the GAS (Goal Attainment Scale), one of our purposes is to monitor the parameters that influence the achievement of functional goals set by patients together with the medical team in order to render the patients as close as possible to achieving their proposed functional goals, thus enhancing their quality of life. By assessing and establishing statistical and clinical correlations between the GAS and quantifiable parameters related to the affected post-stroke upper limb, namely degree of spasticity, motor control, pain level and evolution of pain under treatment with BoNT-A (abobotulinum toxin A), and patients’ overall response to BoNT-A treatment, we aim to quantify the improvement of the therapeutic management of post-stroke patients with spasticity and develop a more personalized and effective approach to their disability and impairment. Results and discussions. The analysis concluded that there were two independent predictors of the Achieved GAS-T score (the study’s endpoint parameter) motor control at any level of the upper limb and number of prior BoNT-A injections. The number of prior BoNT-A injections was an independent predictor of Achieved GAS-T score improvement but had no significant influence over Baseline GAS-T score. Enhancement in proximal and intermediate motor control showed a GAS score improvement of 3.3 points and a 0.93-point GAS score improvement for wrist motor control progress. From a separate viewpoint, patients with motor deficit on the left side have shown significantly greater improvement in Changed GAS-T scores by 2.5 points compared to patients with deficits on the right side; however, we note as a study limitation the fact that there was no statistical analysis over the dominant cerebral hemisphere of each patient. Conclusions. Improvement in the Achieved GAS-T score means better achievement of patients’ goals. Thus, after the BoNT- A intervention, at follow-up evaluation, GAS was found to be directly correlated with improvement in motor control of the affected upper limb. Mobility of the corresponding limb was enhanced by pain decrease during p-ROM (passive range of motion) and by amelioration of spasticity. Materials and Methods. We conducted an observational, non-randomized clinical study on 52 stroke patients, a representative sample of patients with post-stroke spasticity and disability from our neurological rehabilitation clinic, who have been treated and undergone a specific rehabilitation program in our tertiary diagnostic and treatment medical center, including BoNT-A focal treatment for spasticity in the affected upper limb. The primary objective of the study was to assess the influence of abobotulinum toxin A treatment on the Goal Attainment Scale. Secondary objectives of the study included the assessment of BoNT-A treatment efficacy on spasticity with the MAS (Modified Ashworth Scale), pain with the NRS (Numerical Rating Scale), and joint passive range of motion (p-ROM), identifying demographic, clinical, and pharmacological factors that influence the response to BoNT-A treatment, as well as to conduct a descriptive and exploratory analysis of the studied variables.

## 1. Introduction

For each person, establishing certain objectives in everyday life stimulates the adoption of new behaviors, aligns one’s focus, and maintains momentum while also facilitating effective management on the arduous road to improvement [1,2,3]. Goal-setting is a core practice in rehabilitation, characterized as the collaborative process wherein the patient and clinical members of the multidisciplinary team collectively make decisions, informed by thorough discussions, regarding the methods and timing for conducting rehabilitation activities. It aims a clear and comprehensive identification of the underlying reasons for all planned actions [1]. Despite being a fundamental practice, it is not quite a simple challenge for either the patients or the clinicians [2,3]. Ultimately, goal-setting also requires proper measurement [4]. Goal Attainment Scaling (GAS) serves as a technique for assessing the extent of accomplishment in achieving predetermined rehabilitation objectives [5]. The scoring of achievements is performed in a standardized manner, assigning a numerical value that can be subsequently utilized for statistical analysis [2,3,5,6].

Among the leading actors on the pathology stage of neurological rehabilitation is stroke, which stands as one of the primary contributors to adult disability, showcasing a diverse range of clinical signs and symptoms [6], a prominent and pernicious one being increased muscle activity or spasticity as a result of damage to the upper motor neuron (UMN) pathway [7,8,9,10]. Patients with spasticity experience functional impairments [11] stemming from three primary processes: weakness, biomechanical alterations (including soft tissue stiffness, muscle shortening, and tendon contracture), and muscle over-activity resulting from hyperexcitability or the loss of inhibition. 

Extensive scientific evidence now solidly demonstrates the effectiveness of BoNT-A as a focused intervention for reducing spasticity in clinical settings [12,13,14,15,16,17,18,19,20,21]. BoNT-A effectively treats muscle over-activity by blocking the release of acetylcholine at the neuromuscular junction [22]. In stroke patients’ upper limbs, BoNT-A has demonstrated its efficacy in reducing muscle hypertonia, a result expected as muscle over-activity plays a crucial role in increasing muscle tone [23,24,25]. The ability to reduce muscle over-activity is also the basis for increasing p-ROM, although this is, in some regard, unexpected since muscle contracture is typically considered the primary factor leading to decreased p-ROM [26]. Furthermore, there is substantial scientific data indicating that BoNT-A exerts an analgesic effect on the muscles of post-stroke affected limbs [27,28,29,30,31,32] and musculoskeletal disorders [33]. Notwithstanding the beneficial effects of focal spasticity treatment with BoNT-A, we have to bear in mind that it is still a pharmacological intervention that maintains the associated precautions, and, like any other drug, should not be administered unless utterly required, in each patient’s case, in a personalized therapeutic manner. Therefore, current guidelines for the use of BoNT-A in spasticity management [21,34] advocate the adoption of more focused outcome evaluations. These assessments should specifically address the accomplishment of priority goals that bear significant importance for the individual [35].

The rehabilitation medical team is responsible for the therapeutic management of post-stroke patients and, therefore, for the complex management of spasticity. Considering the generous arsenal at our disposal both therapeutically, through the possibility of administering botulinum toxin to combat spasticity, and in terms of accurate assessment through the International Classification of Functioning, Disability and Health (ICF) [36,37] and developed functional scales such as the GAS, we aim to monitor the parameters that influence the achievement of functional goals set by patients together with the medical team [38]. Our ultimate goal is, essentially, that our patients are as close as possible to achieving their proposed functional goals, thus enhancing their quality of life. By assessing and establishing statistical and clinical correlations between the GAS and quantifiable parameters related to the affected upper limb post-stroke, namely degree of spasticity, motor control, pain level and evolution of pain under treatment with BoNT-A, and patients’ overall response to BoNT-A treatment [38], we ultimately aim to develop a more personalized and effective therapeutic management of the disability and impairment of stroke patients.

## 2. Results

Fifty-two patients who met the inclusion criteria participated in the study (Table 1) (age: 56.40 ± 12.66 years; sex: 20 women (38%), 32 men (62%); damaged hemisphere: right 19 (37%) and left 33 (63%); lesion type: ischemia 37 (71%), hemorrhage 15 (29%); time since stroke onset: 16.08 ± 14.00 months; time interval between T0 and T1: 20 days ± 5 days). Among the 52 patients, 28 (51.85%) of them received oral antispasticity medication (baclofen) in a median dose of 42.5 mg per day. 

Of the 52 selected patients, the most frequently injected muscles were the biceps brachii and flexor digitorum superficialis muscles (51 = 98% of patients), followed by the pronator teres muscle (48 = 92% of patients), brachialis muscle (47 = 90% of patients), and flexor digitorum profundus muscle (46 = 88% of patients). The least frequently injected were the flexor carpi ulnaris (14 = 27% of patients) and the flexor pollicis longus (16 = 31% of patients) muscles (see Table 2).

Table 3 shows the univariate simple regression from which we observe the following statistically significant influences: patients with deficits on the left side have a significantly greater improvement in Changed GAS-T scores by 2.5 points compared to patients with deficits on the right side.

Also, patients who received previous treatment with BoNT-A have a higher Changed GAS-T score improvement, with any additional injection resulting in an increase of 0.92 points in the score.

From the table, the following statistically significant influences can be observed: patients who used post-injection splints and/or orthoses have a lower GAS-T score improvement by 2.5 points compared to patients who did not use this adjunctive therapy method.

The difference between discharge and admission FIM influences the GAS-T improvement score; an increase of one point in this difference is associated with an increase of 2.9 points in the Changed GAS-T score (see Table 4). 

From the table of Changed GAS-T score predictors, we can observe several statistically significant relations. First of all, the number of previous injections of BoNT-A at the wrist level positively influences GAS score, with an additional injection being associated with an increase in the Changed GAS-T score of 0.93 points. 

From Table 5, Table 6 and Table 7, it can be observed that the following statistically significant influences exist:-The number of prior injections at the wrist, with an additional injection being associated with an increase in the improved GAS-T score of 0.93 points.-Improved proximal motor control, with an increase of 1 point being associated with an increase in the improved GAS-T score of 3.3 points.-Improved intermediate motor control, with an increase of 1 point being associated with an increase in the improved GAS-T score of 3.3 points.

Consequently, there is a direct proportional relationship between improved proximal motor control and GAS score; in other words, an increase in the force of the muscles of the shoulder girdle of 1 point is associated with an increase in the Changed GAS-T score of 3.3 points. Also, when intermediate motor control is improved, so is the endpoint parameter, with an increase of 1 point being associated with an increase in the Changed GAS-T score of 3.3 points (see Table 5).

Following the above stated statistical methodology, after applying the initial simple univariate linear regression, predictors that did not have statistically significant influence were eliminated from the model via a multiple univariate linear regression, which is presented in Table 8 and uses a backward selection algorithm. 

Thus, the independent predictors improving the Changed GAS-T score, our endpoint parameter, were A. the number of previous BoNT-A injections, which with any additional injection was associated with an increase of 0.72 in the Changed GAS-T score, and B. improved proximal motor control, which with every 1 point increase in the MRC score of the shoulder girdle muscles generated an increase of 2.8 in the Changed GAS-T score. 

The achieved GAS-T score (follow-up evaluation) was assessed in correlation with:-Spasticity decrease across all muscle groups (overall upper limb spasticity reduction) of the affected upper limb;-Overall pain decrease at the affected upper limb (shoulder/elbow/wrist);-Number of prior BoNT-A injections of each patient;-Improvement in motor control of the affected upper limb (proximal, intermediate, distal); composite variables (baseline and follow-up) were employed, quantifying the mentioned parameters at all levels of the affected upper limb.

Significant statistical correlations (*p* < 0.05) were obtained through the use of non-parametric tests, such as Pearson and Spearman, regarding the improvement of the changed GAS-T score with the reduction in upper limb spasticity, motor control enhancement, and pain decrease.

The Spearman’s rank correlation coefficient (Table 9) for the Achieved GAS-T score and overall spasticity of patients’ upper limb is rho = −0.36 *p* = 0.004 (<0.01), indicating a strong negative correlation which indicated that as spasticity increases, the Achieved GAS-T score (final GAS score at follow-up) tends to be lower. The variable MAS2 (composite variable of spasticity evaluated at follow-up) and the Achieved GAS-T score are inversely related.

Table 10 shows the non-parametric correlation for the GAS score between baseline and achieved values in relation to the spasticity decrease and displays a correlation coefficient of statistical significance of rho = 0.052.

Table 11 shows the Pearson correlation coefficient of 0.75 which indicates a very strong correlation between the degree of spasticity before the intervention with the composite variable MAS1 and after the intervention (at follow-up) quantified by the composite variable MAS2.

The Spearman’s rank correlation coefficient rho = −0.33 (*p* = 0.008 < 0.05) indicating a strong negative correlation between pain in the affected upper limb at follow-up (composite variable Pain2) and the achieved GAS-T score and indicating that patients whose level of pain decreases after the intervention have a higher probability of attaining their set functional goals (Table 12).

Table 13 dysplays the Pearson correlation coefficient of 0.84 (=very strong) (*p* < 0.0001) indicates a highly efficient intervention by significantly reducing pain in the affected upper limb. (Pain1 and Pain2 = composite variables consisting of all pain variables at baseline and follow-up moments.)

The Spearman’s rank correlation coefficient of 0.59 (*p* < 0.0001) indicates a highly significant, strong positive correlation. This means that as motor control increases, the final GAS score also increases (Table 14).

The non-parametric 2-tailed *t*-test for Baseline and Achieved GAS and the number of prior BoNT-A treatments is shown in Table 15.

Figure 1 displays the histogram of the achieved GAS-T score distribution, mean, and standard deviation.

The following graphs for Goals 1 (primary) (Figure 2), 2 (Figure 3), and 3 (secondary) (based on Baseline-Figure 4 and Outcome scores-Figure 5)—divided into clusters (=functional domain + upper limb level)—represent the distribution of specific goals the patients had in mind. 

## 3. Discussion

### 3.1. Role of BoNT-A Treatment in GAS 

Being a valuable tool in rehabilitation which measures the extent to which patients are able to achieve their personalized treatment goals, the GAS is not just an assessment tool but also a means of connecting patients with both the medical rehabilitation team and family, friends, and caregivers. Botulinum toxin injections play a crucial role in improving GAS scores in patients with spastic paresis by addressing specific spasticity and spastic-dystonia-related issues hindering goal attainment and by improving the patients’ global functionality [1,2,3,4,5,6,38,39]. Moreover, by reducing muscle spasticity, botulinum toxin injections contribute to enhanced functionality and motor control. This improvement is particularly important for activities such as walking, grasping objects, and other essential daily tasks [12,13,14,15,16,18,19,20,21,22,23,24,25,40,41,42].

The treatment of focal spasticity with botulinum toxin injection in post-stroke spastic patients has proven to be a valuable therapeutic approach. Focal spasticity, affecting specific muscle groups, often contributes to functional limitations in stroke survivors. BoNT-A treatment is of the utmost importance for improvement in active or passive ROM, which in turn enhances the patients’ ability to perform daily activities and improve their overall functional features, with this fact being mirrored in the improvement of the GAS score. Also, BoNT-A plays a crucial role in pain reduction by reducing muscle spasms and promoting muscle relaxation [12,13,14,15,16,18,19,20,21,22,23,24,25,26,27,28,29,30,40,41,43].

A phase IV randomized, controlled placebo trial (RCT) involving multiple centers (*n* = 96) assessed the use of BoNT-A for treating upper limb spasticity post-stroke in Australia. The primary focus was to evaluate the treatment’s impact on quality of life and other person-centered outcomes, including Goal Attainment Scaling (GAS). While the study did not reveal a significant impact on quality of life, as measured by the Assessment of Quality of Life (AQoL) in relation to spasticity reduction, it did indicate a highly significant effect of BoNT-A concerning goal attainment [44]. 

Undoubtedly, we can agree on the abovementioned aspects if we look at our study’s results which attest the importance of BoNT-A treatment in our own subjects, whom, according to the statistical analysis, have had a better chance in achieving their goals, with one reason being BoNT-A treatment. Still, we should not disregard the rehabilitation program or the use of prostethics. As clinicians, it may be quite obvious for us that clinically, patients who comply with the post-stroke rehabilitation program, may have an improvement in the GAS score even without BoNT-A treatment, but the absence of a control group (without BoNT-A treatment administered, even if needed) in our study is based on the ethical aspect that you cannot alter the optimal treatment and deprive the patient of an intervention which is clearly beneficial.

We found that the number of prior injections with BoNT-A in our subjects has a strong and positive correlation with the achieved GAS-T score, but a bit surprisingly, it had no significant statistical correlation with baseline GAS-T score.

### 3.2. Influence of Motor Control over Goal Setting and Achieving

The highlight of the statistical analysis might be the striking correlation between motor control and GAS. With the use of single and multi-variable regression, we have found that each level of the affected upper limb, individually, is a predictor of the GAS score. In other words, improvement in either one of the upper limb levels (shoulder, elbow, wrist) means an improvement in the GAS score, which translates to goal achievement (Table 5, Table 6 and Table 7).

Nevertheless, the relation between motor control, increased muscle tone, spastic dystonia, and decrease in p-ROM is a great strength, so we must not be overjoyed at the idea that a patient exerts good motor control at first, but also exerts negative indicators of functional spasticity [21].

The statistical analysis shows that there are two very important independent predictors for GAS improvement (Table 8): the number of previous BoNT-A injections (i.e., regardless of injection site), which with any additional injection was associated with an increase of 0.72 in the Changed GAS-T score, and improved proximal motor control, which for every 1-point increase in the MRC score of the shoulder girdle muscles generated an increase of 2.8 in the Changed GAS-T score.

### 3.3. Pain Assessment, Management and Impact on GAS

The results of the study regarding pain in affected post-stroke upper limbs are in accordance with the current academic literature; there is no statistical relevance between the GAS and pain score (NRS) at each individual level of shoulder, elbow or wrist, but the overall pain score in the affected upper limb is strongly and negatively correlated with the GAS score improvement. In other words, improvement in the GAS is proportional with the decrease in pain, and therefore, patients are more likely to achieve their goals if the pain management of post-stroke affected upper limb is effective. Bear in mind that in our study, the management of pain in passive joint mobility was not treated with other medication; patients underwent specific rehabilitation program and BoNT-A injection.

### 3.4. Splints and Orthoses as Adjunctive Therapy

Not surprisingly, the study results have shown that the use of splints and orthoses by some of the patients, who most definitely need to use them on a daily basis in order to avoid tendon retractions and deformities, are probably the ones that exhibited higher MAS and NRS scores and also lower FIM and Barthel Index scores and baseline GAS-T scores. Table 16 shows a mild positive correlation between Barthel Index score at hospital admission and use of orthoses. In other words, we should not assume a direct negative correlation of this adjunctive therapeutic method with the constraint of patients’ goals; it is more likely that their appliance is just an indicator of the patients’ overall slow and more restrained functional progress.

However, the use of prosthetics is of great importance in preventing fixed postures, tendon retractions, pain at passive joint mobility, and, ultimately, inability to use the entire limb [45].

### 3.5. Oral Antispasticity Medication

In relation to oral antispasticity drugs, the study solely recorded the administration of baclofen in the subjects. For non-progressive neurological diseases, oral antispasticity medications exhibit only limited effectiveness in treatment [46] and do not include improvement in patients’ quality of life [47]. Additionally, adverse drug reactions were frequently reported [47]. However, amid the various prescribed drugs, baclofen stands out as one of the most effective in clinical practice. An open-label study [48] on oral intake of baclofen observed enhanced motor function and a reduction of almost 90% in spasticity conditions for non-progressive neurological diseases when compared to a control group. A joint study employing a partially blind cross-over with tizanidine and baclofen [49] found no significant difference in the outcome. Nevertheless, these drugs exhibit variations in their side-effect profiles, with baclofen showing fewer serious complications compared to tizanidine [46,47].

Concerning other pharmaceutical classes of antispasticity medication, gabapentin is frequently used, and most of our subjects use it for its neuropathic analgesic effect and to alleviate muscle spasm [49]. Nonetheless, the results of a randomized, placebo-controlled trial [50] have shown no significant changes after administration of gabapentin in patients with regard to spasticity decrease or motor function improvement [46,51]. Other prescribed medications for spasticity are GABA agonists, such as diazepam, clonazepam, alprazolam, lorazepam, midazolam, etc. [46,52], which are also part of our patients’ medication, usually administered inconsistently for their anxiolytic and analgesic effect, their efficacy in spasticity treatment being limited [46,47,51].

Our study did not show a significant statistical correlation between baclofen intake, regardless of the daily dose or lack thereof, and GAS improvement (Table 17).

### 3.6. Study Limitations

One constraint is that we did not inquire about the patients’ dominant brain hemisphere. Severe aphasia served as one of the exclusion criteria, and, given the frequent association of dominant brain lesions with aphasia, we did not deem this information crucial at the time [53].

Furthermore, we note the lack of electromyography or electrical nerve stimulation guidance during BoNT-A injections, which would be of great help in targeting individual NMJs of each muscle, as the clinic does not possess the required devices. All guidance was performed using musculoskeletal 2D ultrasonography.

### 3.7. Future Perspectives

The findings of this study shed a little more light on the intricate role of botulinum toxin (BoNT-A) in facilitating goal achievement for post-stroke patients, particularly in the context of motor control and prior BoNT-A injections. The insights gained contribute to the foundation for future research endeavors aimed at refining and individualizing BoNT-A treatment approaches to enhance goal achievement and functional outcomes for post-stroke patients.

By integrating multidimensional assessments and adopting a personalized approach to rehabilitation, we can strive towards maximizing the therapeutic potential of BoNT-A in optimizing long-term functional recovery and quality of life for individuals affected by stroke.

## 4. Conclusions

Despite the acknowledged limitations, the findings from this study shed light on several key points. The primary clinical aspect that improved the GAS outcome score in post-stroke patients is motor control. Likewise, motor control is greatly correlated with joint mobility, joint or limb pain, and spasticity or spastic dystonia. Present scientific data and multidisciplinary rehabilitation teams diligently address the aforementioned aspects, when necessary and of worth, for the treatment of focal spasticity with BoNT-A. Over the years, the medical community has recognized the numerous advantages of this neural-blocking agent, and there may be even more to uncover. Consequently, BoNT-A has rightfully established itself as a crucial and beneficial ally in achieving therapeutic goals for our patients. As medical specialists, it enables us to treat, care for, and consider the needs of our patients, fostering collaborative efforts to enhance their quality of life.

## 5. Materials and Methods

### 5.1. Study Design

We conducted an observational, non-randomized clinical study on 52 stroke survivors, a representative sample for the population of patients with post-stroke disability who have been treated and undergone a specific rehabilitation program in our tertiary diagnostic and treatment medical center.

All patients received abobotulinumtoxin A (Dysport^®^, Ipsen, Paris, France) because the hospital where the study was conducted is supplied with that type of toxin.

The primary objective of the study was to evaluate the influence of BoNT-A treatment on the Goal Attainment Scale. Secondary objectives of the study included assessment of BoNT-A treatment efficacy; identifying demographic, clinical, and pharmacological factors that influence the response to BoNT-A treatment; as well as conducting a descriptive analysis of the studied variables.

BoNT-A was administered under ultrasound guidance. The selection of muscles and BoNT-A doses to be injected were individually determined based on the clinical presentation, aiming to reduce hypertonia, improve motor control and passive joint mobility, and alleviate related disability and pain. Following BoNT-A treatment, all patients underwent 10 days of a standardized inpatient rehabilitation program in our clinic [33,54,55], consisting of physical therapy and electrotherapy twice per day, and as an adjunctive therapy method, the use of splints and/or orthoses for spastic limb posture, which is included as a variable in the database.

The primary endpoint of the study was the difference between GAS-T score at T0 (Baseline GAS-T score)—assessment at the point of inpatient hospital admission—and T1 (Achieved GAS-T score), which represented the follow-up evaluation at 20 days (±5 days) after hospital discharge.

Following the development protocol, we conducted a retrospective review of the medical records of patients who had undergone BoNT-A injections for the treatment of muscle hypertonia in the pectoralis major muscle and flexor muscles of the elbow, wrist, and fingers. Clinical assessment also included motor control, passive joint mobility, and pain at the shoulder, elbow, wrist, and finger level. The data collection period spanned from January 2019 to December 2023. To maintain consistency and reliability, all assessments were performed by the same examiner.

The decision to solely focus on the pectoralis major muscle for the shoulder level was based on the observation that our patients, typically, had the other muscles acting on the shoulder joint infiltrated only in a minority of cases, while the majority of patients with hypertonic upper limbs received injections in the above-mentioned pectoralis major muscles and the elbow, wrist, and finger flexors. Moreover, shoulder pain is a common issue in patients with upper limb hypertonia, and existing publications on the effects of BoNT-A on pain in stroke patients often concentrate on shoulder pain [56]; by far, the most important, most frequently injected, and most relevant for decreasing muscle pain acting on the shoulder joint is the pectoralis major muscle [57,58,59].

This study adhered to ethical standards outlined in the World Medical Association’s Code of Ethics (Declaration of Helsinki) for experiments involving humans. Written informed consent was obtained from all participants and the study received approval from the local ethics committee (Ethics Committee on Human Research of INRMFB, approval code 1/07.01.2019, approval date 7 January 2019).

### 5.2. Patient Selection

#### 5.2.1. Inclusion Criteria

Patients who have fulfilled the written informed consent;Age ≥ 18 and ≤80 years;Hemiparesis due to a single stroke occurred ≥2 months before the assessment;Presence of muscle hypertonia of shoulder, elbow, wrist, and/or finger level;Clinical assessment performed just before (T0 = baseline, at the moment of inpatient hospital admission) and after BoNT-A treatment (T1 = follow-up evaluation at 20 days ± 5 days after hospital discharge), which included: (a) motor control at shoulder, elbow, wrist, and fingers levels; (b) pain perceived in shoulder, elbow, wrist, and fingers during passive mobilization; (c) muscle tone of pectoralis major, elbow, wrist and finger flexors; and (d) goal setting and GAS assessment.

#### 5.2.2. Exclusion Criteria

Recurrent strokes or other medical conditions in addition to stroke likely to interfere with the clinical assessment reported in the inclusion criteria:Use of intrathecal baclofen [60];BoNT-A or other neural-blocking agents [61,62,63] injected in the upper limb three months before assessment at T0;Patients who experienced adverse effects from previous BoNT-A injections (e.g., myalgia, muscle weakness, asthenia, flu-like syndrome, local reactions at the injection site, etc.);Severe cognitive impairment;Severe aphasia interfering with patient’s assessment;Degree of spasticity <= 1 or 4 on MAS (Modified Ashworth Scale);Patients who have refused the written informed consent.

Regarding previous botulinum toxin injections, patients with prior botulinum toxin treatment were injected only after three or more months after the last injection in order to avoid a potential cumulative effect and to avoid false-positive results.

### 5.3. Intramuscular Diffusion of BoNT-A

Abobotulinumtoxin A displays pharmacological effects by inhibiting the release of acetylcholine from the pre-synaptic nerve terminal, thereby obstructing peripheral cholinergic transmission at the neuromuscular junction (NMJ). This leads to a decrease in muscle contraction and a reversible reduction in muscle power, which is dependent on the dosage. BoNT-A is more actively absorbed by active NMJs compared to those at rest. Additionally, BoNT-A blocks gamma-efferent fiber NMJs in muscle spindles, likely contributing to a reduction in reflex sensitivity. BoNT-A is administered through intramuscular injections into specific muscles. While it can diffuse across muscle fascial barriers, its impact is primarily concentrated in the injected muscle. BoNT-A exerts a certain degree of diffusion within the muscle; nevertheless, injections positioned near motor end-plate zones may potentially yield greater efficiency [41,42,43,53,64].

### 5.4. GAS Assessment

In order to perform the GAS assessment, we took into account the level of accomplishment in attaining three predetermined rehabilitation objectives: a primary goal and two secondary goals as a means to determine the final changed GAS-T score, which was thus considered the ultimate endpoint evaluation score. As to facilitate statistical correlations of the overall pool of individual goals attested, the adopted protocol analyzed all the goals and comprised them by means of the basic parameter which de facto influenced them: motor control, joint mobility, and joint pain at each of the levels mentioned before (shoulder, elbow, wrist). Accordingly, we also found it necessary to introduce the hand-finger level for a better assessment of ability and pain at this level with regard to the few patients who exhibited both good hand-finger motor control and mobility at baseline evaluation (T0). Hence, our simplified versions of the pooled objectives were eventually broken down to the following parameters:

Proximal (shoulder) or intermediate (elbow) or distal (wrist) level:1.a.Motor control;1.b.Joint pain;1.c.Joint mobility.

Utmost distal level (hand-fingers): hand ability (which, in essence, represents the comprised motor control and mobility parameters, and is referred to the few patients with a higher overall functionality at this level, in order to thoroughly quantify their outcome).

### 5.5. Muscle Tone Assessment

The evaluation of the tone in the elbow and finger flexors was conducted using the Modified Ashworth Scale (MAS) [65], a 6-point scale that spans from 0 (indicating no increase in tone) to 4 (representing limb rigidity in either flexion or extension) [66].

MAS was designed to enhance the sensitivity of the original scale. An additional category, 1+, was introduced between 1 and 2 to differentiate among patients, particularly those positioned at the lower end of the scale, expanding the original Ashworth scale to six classifications instead of five, as outlined below:

0: No increase in muscle tone.

1: Slight increase in muscle tone, noted by a catch during flexion or extension movement.

1+: Slight increase in muscle tone, characterized by a catch followed by minimal resistance throughout the range of motion (ROM).

2: More marked increase in muscle tone, making limb flexion more challenging but still achievable.

3: Considerable increase in muscle tone, making passive movement difficult.

4: Limb is rigid in either flexion or extension.

In our study, for the purpose of simplifying the analysis of MAS scores, occurrences of 1+ were modified and assigned a value of 1.5 [52].

### 5.6. Pain Assessment

Pain was assessed at shoulder, elbow, and wrist level during passive mobilization and rated on the Numerical Rating Scale (pain-NRS) that ranged from 0 to 10 [67,68,69].

### 5.7. Motor Control Assessment

To evaluate the motor control at proximal, intermediate, and distal levels, we used the MRC power muscle scale [70] to quantify the extent to which motor control has changed at each level at T1 (follow-up evaluation) in comparison with T0 (baseline).

### 5.8. Statistical Analysis

Regarding statistical methodology, a single-variable linear regression was initially used, followed by a multiple-variable regression. The dependent variable in this analysis was the Changed GAS-T score, which represents the difference between the Baseline GAS-T score and Achieved GAS-T score but not as a subtraction (as used with the other variables in order to quantify the difference between baseline and follow-up scores), but in the form of a complex mathematical function specifically developed to appraise overall changes to the Goal Attainment Scale. It can be calculated via the following formula or with the downloadable Excel spreadsheet [44]:$$T=50+\frac{10\sum w_{i}x_{i}}{\sqrt{\left(1-\rho \right)\sum w_{i}^{2}+\rho {\left(\sum w_{i}\right)}^{2}}}$$

Accordingly, the analyzed independent variables included various demographic, clinical, and pharmacological aspects monitored in our study.

The results are presented as the mean (standard deviation) for continuous or ordinal variables and as absolute frequency (percentage) for categorical variables. Baseline characteristics of the subjects included the assessment of pain, motor control, and spasticity (MAS score) at shoulder, elbow, wrist, and fingers and were reported both collectively and separately.

The statistical analysis was performed using the Statistical Package for the Social Sciences (SPSS), version [Amos v20], to assess the relationships between variables and derive meaningful insights. Also, the R software was used (version 4.3.2: Copyrigh© 2023 The R Foundation for Statistical Computing, R Core Team (2023); R: A language and environment for statistical computing; R Foundation for Statistical Computing, Vienna, Austria) with the following packages: gtsummary and sjPlot. The significance level (alpha) of the study was set at 0.05, so *p*-values less than 0.05 were considered statistically significant.

## Figures and Tables

**Figure 1 toxins-16-00172-f001:**
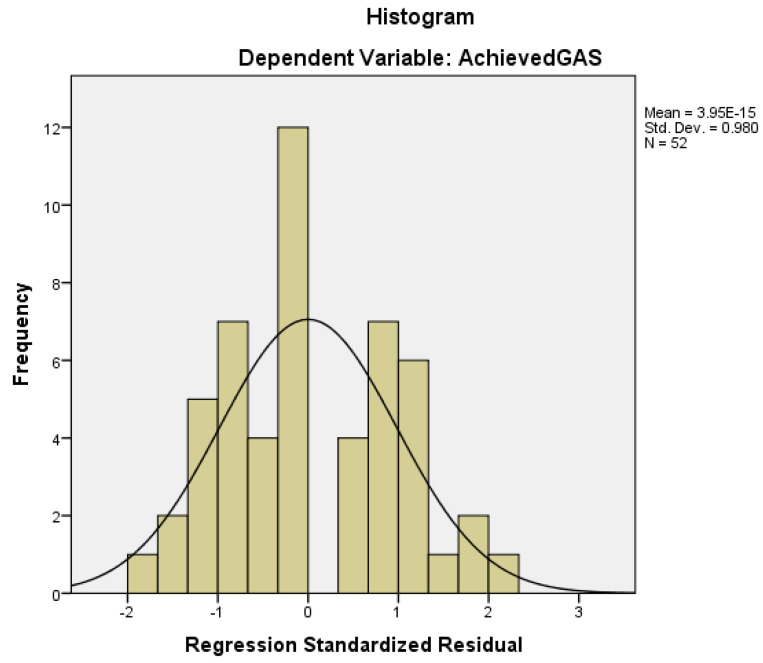
Histogram of the Achieved GAS-T score distribution, mean, and standard deviation.

**Figure 2 toxins-16-00172-f002:**
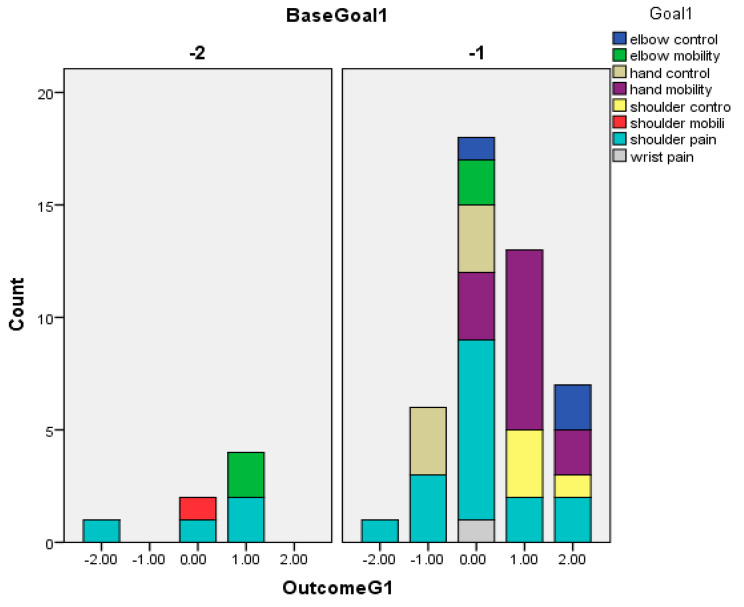
Outcome of patients’ primary goals (goal 1) in relation to GAS.

**Figure 3 toxins-16-00172-f003:**
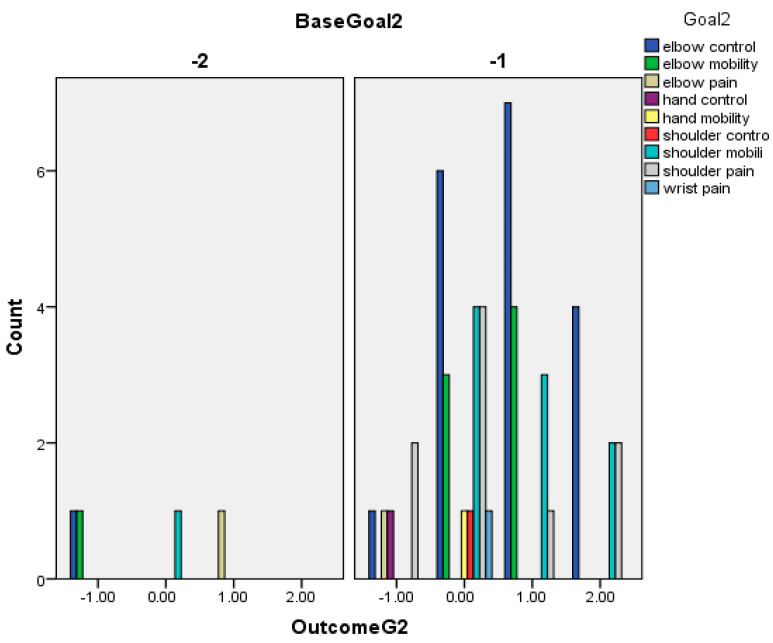
Outcome of patients’ secondary goals (goal 2) in relation to GAS.

**Figure 4 toxins-16-00172-f004:**
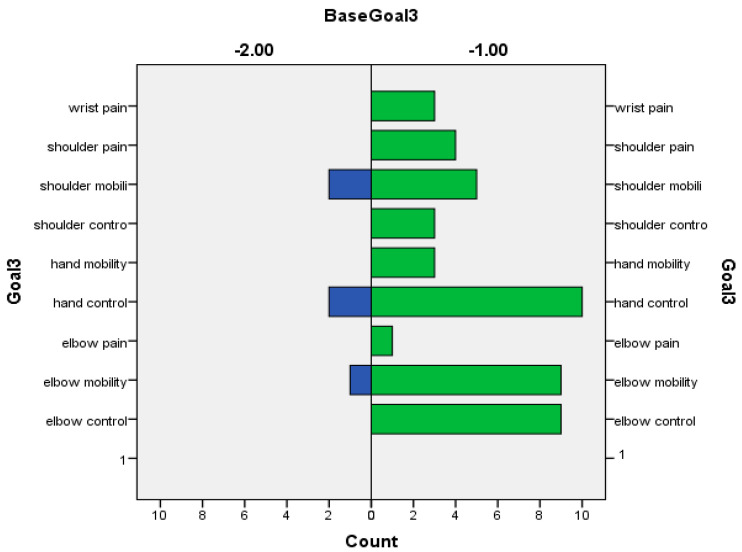
Baseline values of patients’ secondary goals (goal 3) in relation to GAS.

**Figure 5 toxins-16-00172-f005:**
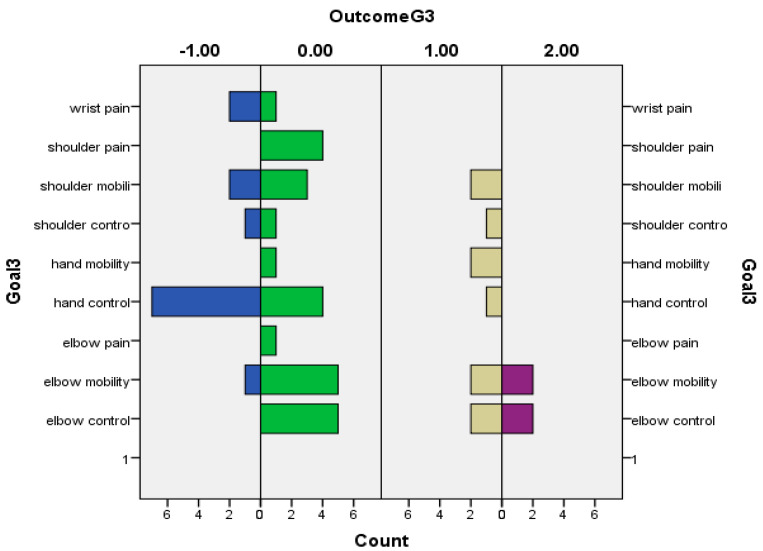
Outcome of patients’ secondary goals (goal 3) in relation to GAS.

**Table 1 toxins-16-00172-t001:** Descriptive statistical analysis of variables.

Variable	N = 52
Sex, *n* (%)	
F	20 (38)
M	32 (62)
Age, Mean (SD)	56.40 (12.66)
Stroke etiology, *n* (%)	
Hemorrhage	15 (29)
Ischemia	37 (71)
Time since stroke onset, Mean (SD)	16.08 (14.00) months
Damaged hemisphere, *n* (%)	
Right	19 (37)
Left	33 (63)
Oral antispastic treatment (baclofenum), *n* (%)	
Yes	28 (54)
No	24 (46)
Prior Nr. of BoNT-A injections (i.e.), Mean (SD)	1.46 (1.97)
Splints/orthoses post-injection, *n* (%)	
Yes	26 (51)
No	26 (51)
FIM score admission, Mean (SD)	5.04 (1.49)
FIM score follow-up, Mean (SD)	5.23 (1.57)
Barthel score admission, Mean (SD)	75.48 (21.13)
Barthel score follow-up, Mean (SD)	76.92 (21.42)
Pectoralis major BoNT-A dose (N = 39), Mean (SD)	130.51 (48.77)
MAS Pectoralis major T0, Mean (SD)	1.72 (0.85)
MAS Pectoralis major T1, Mean (SD)	0.94 (0.52)
Prior Nr. BoNT-A injections shoulder, Mean (SD)	0.96 (1.56)
NRS score shoulder T0, Mean (SD)	3.04 (1.90)
NRS score shoulder T1, Mean (SD)	1.50 (1.31)
Biceps brachii BoNT-A dose, Mean (SD)	139.02 (59.90)
MAS Biceps brachii T0, Mean (SD)	2.02 (0.54)
MAS Biceps brachii T1, Mean (SD)	1.08 (0.48)
Brachialis BoNT-A dose (N = 47), Mean (SD)	111.70 (45.70)
MAS Brachialis T0, Mean (SD)	1.82 (0.70)
MAS Brachialis T1, Mean (SD)	0.97 (0.51)
Brachioradialis BoNT-A dose (N = 40), Mean (SD)	105.00 (38.89)
MAS Brachioradialis T0, Mean (SD)	1.65 (0.79)
MAS Brachioradialis T1, Mean (SD)	0.91 (0.56)
Pronator teres BoNT-A dose (N = 48), Mean (SD)	119.58 (38.41)
MAS Pronator teres T0, Mean (SD)	1.92 (0.62)
MAS Pronator teres T1, Mean (SD)	1.02 (0.53)
Prior Nr. BoNT-A injections elbow, Mean (SD)	1.42 (1.92)
NRS score elbow T0, Mean (SD)	1.15 (1.38)
NRS score elbow T1, Mean (SD)	0.60 (0.82)
Flexor digitorum superficialis BoNT-A dose (N = 51), Mean (SD)	128.53 (45.36)
MAS Flexor digitorum superficialis T0, Mean (SD)	2.16 (0.56)
MAS Flexor digitorum superficialis T1, Mean (SD)	1.18 (0.46)
Flexor digitorum profundus BoNT-A dose (N = 46), Mean (SD)	103.48 (36.30)
MAS Flexor digitorum profundus T0, Mean (SD)	1.86 (0.59)
MAS Flexor digitorum profundus T1, Mean (SD)	1.07 (0.46)
Flexor carpi radialis BoNT-A dose (N = 26), Mean (SD)	117.88 (27.50)
MAS Flexor carpi radialis T0, Mean (SD)	1.45 (1.00)
MAS Flexor carpi radialis T1, Mean (SD)	0.99 (0.53)
Flexor carpi ulnaris BoNT-A dose (N = 14), Mean (SD)	99.29 (25.86)
MAS Flexor carpi ulnaris T0, Mean (SD)	0.85 (0.90)
MAS Flexor carpi ulnaris T1, Mean (SD)	0.56 (0.60)
Prior Nr. BoNT-A injections wrist, Mean (SD)	1.37 (1.85)
NRS score wrist T0, Mean (SD)	0.92 (1.37)
NRS score wrist T1, Mean (SD)	0.52 (0.78)
Flexor pollicis longus BoNT-A dose (N = 16), Mean (SD)	79.69 (24.53)
MAS Flexor pollicis longus T0, Mean (SD)	0.88 (1.06)
MAS Flexor pollicis longus T1, Mean (SD)	0.48 (0.54)
Improved proximal motor control, Mean (SD)	0.69 (0.64)
Improved intermediate motor control, Mean (SD)	1.00 (0.66)
Improved distal motor control, Mean (SD)	0.62 (0.60)
Outcome of Changed GAS-T score, Mean (SD)	19.33 (3.73)
FIM score dif., Mean (SD)	0.19 (0.40)
Barthel score dif., Mean (SD)	1.44 (3.62)
NRS shoulder score dif., Mean (SD)	1.54 (1.26)
NRS elbow score dif., Mean (SD)	0.56 (0.83)
NRS wrist score dif., Mean (SD)	0.40 (0.75)
MAS Pectoralis major score dif., Mean (SD)	0.78 (0.61)
MAS Biceps brachii score dif., Mean (SD)	0.94 (0.43)
MAS Brachialis score dif., Mean (SD)	0.85 (0.48)
MAS Pronator teres score dif., Mean (SD)	0.90 (0.42)
MAS Flexor digitorum superficialis score dif., Mean (SD)	1.10 (0.51)
MAS Flexor carpi radialis score dif., Mean (SD)	0.46 (0.73)
MAS Flexor carpi ulnaris score dif., Mean (SD)	0.29 (0.55)
MAS Flexor pollicis longus score dif., Mean (SD)	0.39 (0.66)

**Table 2 toxins-16-00172-t002:** Muscles injected and doses of BoNT-A (abobotulinumtoxin A) for each muscle.

Muscle	Number of Injected Patients (Percentage)	Dose of BoNT-A (UI): Mean ± SD
Shoulder muscles		
Pectoralis major	39/52 (75%)	130.51 ± 48.77
Elbow flexor muscles		
Biceps brachii	51/52 (98%)	139.02 ± 59.90
Brachialis	47/52 (90%)	111.70 ± 45.70
Brachioradialis	40/52 (77%)	105.00 ± 38.89
Pronator teres	48/52 (92%)	119.58 ± 38.41
Wrist flexor muscles		
Flexor carpi radialis	26/52 (50%)	117.88 ± 27.50
Flexor carpi ulnaris	14/52 (27%)	99.29 ± 25.86
Finger flexor muscles		
Flexor digitorum superficialis	51/52 (98%)	128.53 ± 45.36
Flexor digitorum profundus	46/52 (88%)	103.48 ± 36.30
Flexor pollicis longus	16/52 (31%)	79.69 ± 24.53

**Table 3 toxins-16-00172-t003:** Demographic variables as predictors of Changed GAS-T score.

Predictors	N	Beta (95% CI) ^1^	*p*-Value
Sex			
F	20	—	—
M	22	−0.81 (−2.9 to 1.3)	0.45
Age	—	−0.04 (−0.13 to 0.04)	0.29
Stroke etiology			
Hemorrhagic	15	—	—
Ischemic	37	−1.3 (−3.6 to 0.89)	0.24
Time since stroke onset	—	0.06 (−0.01 to 0.13)	0.12
Damaged hemisphere			
Left	19	—	—
Right	33	2.5 (0.51 to 4.5)	0.017
Oral myorelaxant treatment			
Yes	28	—	—
No	24	−1.9 (−3.9 to 0.10)	0.068
Non-BoNT-naïve patients	27	0.92 (0.46 to 1.4)	<0.001

^1^ CI = Confidence Interval.

**Table 4 toxins-16-00172-t004:** General variables as predictors of Changed GAS-T score.

Predictors	N	Beta (95% CI) ^1^	*p*-Value
Splints/orthoses			
Yes	26	—	—
No	26	−2.5 (−4.4 to −0.55)	0.015
FIM score dif. ^2^	—	2.9 (0.46 to 5.4)	0.024
Barthel score dif. ^3^	—	0.18 (−0.10 to 0.46)	0.21
Nr. of prior BoNT-A injections (i.e.,)	25	0.92 (0.46 to 1.4)	<0.001

^1^ CI = Confidence Interval. ^2^ Difference in FIM scale = FIM score at follow-up (T1)—FIM score at hospital admission (T0). ^3^ Difference in Barthel scale = Barthel score at follow-up (T1)—Barthel score at hospital admission (T0).

**Table 5 toxins-16-00172-t005:** Shoulder variables as predictors of Changed GAS-T score.

Predictors	N	Beta (95% CI) ^1^	*p*-Value
NRS shoulder score dif. ^2^	—	0.72 (−0.08 to 1.5)	0.082
Nr. of prior BoNT-A injections Pectoralis major	—	0.61 (−0.03 to 1.3)	0.068
Improvement in proximal motor control ^3^	—	3.3 (2.0 to 4.6)	<0.001
MAS Pectoralis major score dif. ^4^	—	1.1 (−0.54 to 2.8)	0.19
Pectoralis major BoNT-A dose	—	0.01 (−0.01 to 0.02)	0.46

^1^ CI = Confidence Interval. ^2^ NRS shoulder score difference = NRS shoulder score at T1—NRS shoulder score at T0. ^3^ Improvement in proximal motor control = MRC score at T1—MRC score at T0 for shoulder girdle muscles (i.e.,). ^4^ MAS Pectoralis major score difference = MAS Pectoralis major score at T1—MAS Pectoralis major score at T0.

**Table 6 toxins-16-00172-t006:** Elbow variables as predictors of Changed GAS-T score.

Predictors	N	Beta (95% CI) ^1^	*p*-Value
Nr. of prior BoNT-A injections for elbow flexor muscles (i.e.,)	—	0.91 (0.44 to 1.4)	<0.001
MAS Biceps brachii score dif. ^2^	—	1.4 (−1.05 to 3.83)	0.26
MAS Brachialis score dif. ^2^	—	1.36 (−0.8 to 3.53)	0.21
MAS Brachioradialis score dif. ^2^	—	0.01 (−0.01 to 0.03)	0.17
MAS Pronator teres score dif. ^2^	—	1.8 (−0.61 to 4.2)	0.15
NRS elbow score dif. ^3^	—	0.9 (−0.08 to 1.8)	0.09
Biceps brachii BoNT-A dose	—	0.00 (−0.02 to 0.02)	0.99
Brachialis BoNT-A dose	—	0.00 (−0.02 to 0.02)	0.74
Brachioradialis BoNT-A dose	—	0.01 (−0.01 to 0.03)	0.17
Pronator teres BoNT-A dose	—	0.01 (−0.01 to 0.03)	0.31
Improvement in intermediate motor control ^4^	—	3.1 (1.8 to 4.4)	<0.001

^1^ CI = Confidence Interval. ^2^ MAS score difference = MAS score at T1—MAS score at T0 (for each of the elbow flexor muscles). ^3^ NRS elbow score difference = NRS elbow score at T1—NRS elbow score at T0. ^4^ Improvement in intermediate motor control = MRC score at T1—MRC score at T0 for elbow extensor muscles (i.e.,).

**Table 7 toxins-16-00172-t007:** Wrist and finger variables as predictors of Changed GAS-T score.

Predictors	N	Beta (95% CI) ^1^	*p*-Value
Nr. of prior BoNT-A injections for wrist/fingers flexor muscles (i.e.,)	—	0.93 (0.43 to 1.4)	<0.001
NRS wrist score dif. ^2^	—	−0.51 (−1.9 to 0.87)	0.47
Flexor digitorum superficialis BoNT-A dose	—	0.00 (−0.03 to 0.02)	0.65
Flexor digitorum profundus BoNT-A dose	—	0.01 (−0.01 to 0.03)	0.54
Flexor carpi radialis BoNT-A dose	—	0.00 (−0.02 to 0.01)	0.83
Flexor carpi ulnaris BoNT-A dose	—	0.00 (−0.02 to 0.02)	0.93
Flexor pollicis longus BoNT-A dose	—	0.02 (−0.01 to 0.04)	0.22
MAS score dif. Flexor digitorum superficialis ^3^	—	0.52 (−1.5 to 2.6)	0.62
MAS score dif. Flexor digitorum Profundus ^3^	—	0.3 (−0.6 to 0.3)	0.09
MAS score dif. Flexor carpi radialis ^3^	—	0.00 (−1.4 to 1.4)	0.99
MAS score dif. Flexor carpi ulnaris ^3^	—	0.21 (−1.7 to 2.1)	0.83
MAS score dif. Flexor pollicis longus ^3^	—	1.3 (−0.25 to 2.8)	0.11
Improvement in distal motor control ^4^	—	2.4 (0.85 to 4.0)	0.004

^1^ CI = Confidence Interval. ^2^ NRS wrist score difference = NRS wrist score at T1—NRS wrist score at T0. ^3^ MAS score difference = MAS score at T1—MAS score at T0 (for each of the wrist/fingers flexor muscles). ^4^ Improvement in distal motor control = MRC score at T1—MRC score at T0 for wrist/fingers extensor muscles.

**Table 8 toxins-16-00172-t008:** Independent predictors for GAS improvement.

Predictors	N	Beta (95% CI) ^1^	*p*-Value
A. Nr. of prior BoNT-A injections (i.e.,)	0.72	0.73 (0.31 to 1.1)	0.001
B. Improvement in proximal motor control ^2^	2.8	2.8 (1.6 to 4.0)	<0.001

^1^ CI = Confidence Interval. ^2^ Improvement in proximal motor control = MRC score at T1—MRC score at T0 for shoulder girdle muscles.

**Table 9 toxins-16-00172-t009:** Non-parametric correlation for GAS score and spasticity degree.

			MAS2	AchievedGAS
Spearman’s rho	MAS2	Correlation Coefficient	1.000	−0.362 **
		Sig. (1-tailed)	.	0.004
		N	52	52
	AchievedGAS	Correlation Coefficient	−0.362 **	1.000
		Sig. (1-tailed)	0.004	.
		N	52	52

** Correlation is significant at the 0.01 level (1-tailed).

**Table 10 toxins-16-00172-t010:** Non-parametric correlation for GAS score (baseline and achieved) and spasticity degree.

			MAS2	AchievedGAS	BaselineGAS
Spearman’s rho	MAS2	Correlation Coefficient	1.000	−0.362 **	−0.320 *
		Sig. (1-tailed)	.	0.004	0.010
		N	52	52	52
	AchievedGAS	Correlation Coefficient	−0.362 **	1.000	0.228
		Sig. (1-tailed)	0.004	.	0.052
		N	52	52	52
	BaselineGAS	Correlation Coefficient	−0.320 *	0.228	1.000
		Sig. (1-tailed)	0.010	0.052	.
		N	52	52	52

** Correlation is significant at the 0.01 level (1-tailed). * Correlation is significant at the 0.05 level (1-tailed).

**Table 11 toxins-16-00172-t011:** Pearson correlation between MAS composite variables.

		MAS1	MAS2
MAS1	Pearson Correlation	1	0.745 **
	Sig. (1-tailed)		0.000
	N	52	52
MAS2	Pearson Correlation	0.745 **	1
	Sig. (1-tailed)	0.000	
	N	52	52

** Correlation is significant at the 0.01 level (1-tailed).

**Table 12 toxins-16-00172-t012:** Non-parametric correlation between GAS and pain.

			AchievedGAS	Pain2
Spearman’s rho	AchievedGAS	Correlation Coefficient	1.000	−0.333 **
		Sig. (1-tailed)	.	0.008
		N	52	52
	Pain2	Correlation Coefficient	−0.333 **	1.000
		Sig. (1-tailed)	0.008	.
		N	52	52

** Correlation is significant at the 0.01 level (1-tailed).

**Table 13 toxins-16-00172-t013:** Pearson correlation between global pain level at baseline and outcome.

		Pain1	Pain2
Pain1	Pearson Correlation	1	0.843 **
	Sig. (1-tailed)		0.000
	N	52	52
Pain2	Pearson Correlation	0.843 **	1
	Sig. (1-tailed)	0.000	
	N	52	52

** Correlation is significant at the 0.01 level (1-tailed).

**Table 14 toxins-16-00172-t014:** Correlation between motor control improvement and GAS.

			Motor Control Global Upper Limb Increase	Achieved GAS-T
Spearman’s rho	Motor Control Global Upper Limb Increase	Correlation Coefficient	1.000	0.591 **
		Sig. (1-tailed)	.	0.000
		N	52	52
	Achieved GAS-T	Correlation Coefficient	0.591 **	1.000
		Sig. (1-tailed)	0.000	.
		N	52	52

** Correlation is significant at the 0.01 level (1-tailed).

**Table 15 toxins-16-00172-t015:** Non-parametric 2-tailed *t*-test for Baseline and Achieved GAS and number of prior BoNT-A treatments.

			PriorBTinj	BaselineGAS	AchievedGAS
Spearman’s rho	PriorBTinj	Correlation Coefficient	1.000	−0.002	0.453 **
		Sig. (2-tailed)	.	0.988	0.001
		N	52	52	52
	BaselineGAS	Correlation Coefficient	−0.002	1.000	0.228
		Sig. (2-tailed)	0.988	.	0.104
		N	52	52	52
	AchievedGAS	Correlation Coefficient	0.453 **	0.228	1.000
		Sig. (2-tailed)	0.001	0.104	.
		N	52	52	52

** Correlation is significant at the 0.01 level (2-tailed).

**Table 16 toxins-16-00172-t016:** Non-parametric test for Barthel Index and use of orthoses or splints.

			Orthos	Barthel1
Spearman’s rho	orthos	Correlation Coefficient	1.000	0.220 *
		Sig. (1-tailed)	.	0.049
		N	52	52
	barthel1	Correlation Coefficient	0.220 *	1.000
		Sig. (1-tailed)	0.049	.
		N	52	52

* Correlation is significant at the 0.05 level (1-tailed).

**Table 17 toxins-16-00172-t017:** Pearson correlation for baclofen treatment and GAS.

		Baclofen	AchievedGAS
baclofen	Pearson Correlation	1	−0.111
	Sig. (2-tailed)		0.434
	N	52	52
AchievedGAS	Pearson Correlation	−0.111	1
	Sig. (2-tailed)	0.434	
	N	52	52

No statistical significance.

## Data Availability

Data provided upon reasonable request.

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
