# Peer review of "The Complex Role of Botulinum Toxin in Enhancing Goal Achievement for Post-Stroke Patients"

_toxins, 2024, doi:10.3390/toxins16040172_

Round 1
Reviewer 1 Report
Comments and Suggestions for Authors
This is a well written study investigates the complex relationship between the treatment of post-stroke spasticity using botulinum toxin and the attainment of patients' objectives. Factors such as motor control and the frequency of prior BoNT-A treatment serve as individual predictors of improvement in Goal Attainment Scaling (GAS). As a result, these findings hold importance for healthcare professionals, including physicians and physiotherapists.
However, before it needs refined and more description of using botulinum neurotoxin into these patients. First of all, please discuss intramuscular neural distribution of spasicity in the discussion. Since the spasticity needs a lots of doses of botulinum toxin, with targeting multiple muscles they should be accurately injected with small dose of the amount.
Please refer and cite these and discuss about it in the discussion. Such as "Distribution of the intramuscular innervation of the triceps brachii: Clinical importance in the treatment of spasticity with botulinum neurotoxin".
Overall, the paper seems to be helpful for the readers.
Reviewer 2 Report
Comments and Suggestions for Authors
After a thorough review of the manuscript titled "The Complex Role of Botulinum Toxin in Enhancing Goal Achievement for Post-Stroke Patients," I find it to be a well-constructed observational, non-randomized clinical study. The study meticulously presents participant information, assessments, therapeutic interventions, and a comprehensive 20-week follow-up period, shedding light on the impact of Botulinum Toxin on Goal Attainment Scaling—a subject of considerable interest yet lacking comprehensive exploration despite its extensive use across medical specialties.
While commendable, there are several points that require attention and clarification:
1. It would be beneficial to mention the dominant side of the brain, if possible, particularly in consideration of the patient's handedness prior to hemiplegia onset and the location of the lesion. Discussing these factors could provide valuable insights into the study findings.
2. The choice of a 20-week follow-up period is understandable given the prolonged action of Botulinum Toxin. However, considering that its effect may not persist beyond three months in many patients, clarification on how this variability was addressed, particularly in cases of previous toxin use, is warranted.
3. Assuming Dysport 500U was utilized based on the reported units, it is essential to explicitly state this in the manuscript and confirm whether the same toxin was administered to all patients.
4. In relation to the conversion of MAS scale values, it's worth highlighting that converting 1+ to 1.5 for statistical analysis deviates from the more conventional practice of converting to a 0-5 scale, although the utilization of 1.5 is permissible. Strengthening the credibility of the analysis could be achieved by referencing relevant literature supporting this particular approach.
5. The average time since stroke onset, spanning 16 months, appears lengthy given contemporary guidelines advocating early Botulinum Toxin use. Moreover, the reported average number of previous toxin administrations raises questions regarding the timing and justification of these interventions.
6. The utilization of botulinum toxin in previous treatments correlates with improved GAS-T scores. This phenomenon could potentially be attributed to the cumulative effects of toxin administration, optimized unit distribution by physicians with prior experience, and enhanced patient engagement in rehabilitation programs. It is suggested to explore relevant literature references and further elaborate on this observation in the discussion section.
In conclusion, while acknowledging the authors' commendable efforts, I recommend a thorough revision addressing the aforementioned points to strengthen the manuscript's clarity and scientific rigor.
Round 2
Reviewer 2 Report
Comments and Suggestions for Authors
The manuscript has been sufficiently improved to warrant publication in Toxins.
Author Response
Thank you very much for your comments, advices and approval of our research.
Best regards from our medical and research team.